# Pilot-scale depuration demonstrates the suitability of non-pathogenic *Vibrio parahaemolyticus* as a surrogate for commercial-scale validation studies

Spencer L. Lunda [1]*, Samantha Kilgore[2], Jennifer M. Hesser[1], Joy G. Waite-Cusic[2], Carla B. Schubiger[1,3]

**1** Department of Biomedical Sciences, Carlson College of Veterinary Medicine, Oregon State University, Corvallis, Oregon, United States of America, **2** Department of Food Science and Technology, Oregon State University, Corvallis, Oregon, United States of America, **3** Aquarium Science Program, Oregon Coast Community College, Newport, Oregon, United States of America

* Lundas@oregonstate.edu

## Abstract

Shellfish producers have proposed using depuration systems to reduce *Vibrio parahaemolyticus* levels by supporting the oysters' natural purging ability. For commercial viability, depuration efficacy must be validated in relevant species and at scale using appropriate conditions, including time and temperature. This study aims to I) compare depuration efficacy across oyster species, II) assess non-pathogenic *V. parahaemolyticus* as a surrogate, and III) optimize pilot-scale depuration to identify process variables likely to achieve >3.0-log reduction of *V. parahaemolyticus* at commercial-scale. Three oyster species (*Crassostrea gigas*, *Crassostrea sikamea*, *Crassostrea virginica*) were exposed to artificial seawater (35 ppt) containing five non-pathogenic or five pathogenic *V. parahaemolyticus* strains. Inoculated oysters were placed in a pilot-scale recirculating depuration system at 5, 11, or 13 °C for 7 days. Three replicate depuration trials were conducted, sampling five oysters per species and the two different bacterial inoculations every 24 hours. *V. parahaemolyticus* in oyster tissue was enumerated using serial dilutions and spread plating techniques on Thiosulfate-Citrate-Bile Salts-Sucrose agar. In *C. gigas* and *C. sikamea*, inoculation achieved at least a 5 log CFU/g *V. parahaemolyticus*, but *C. virginica* did not consistently reach target density within 24 hours. Accumulation of non-pathogenic and pathogenic strains was similar across species. Depuration at 11°C achieved a > 3 log CFU/mL reduction of the pathogenic strain combination in *C. gigas* and *C. sikamea* tissues within five days, while *C. virginica* averaged 2.8 log CFU/mL. *V. parahaemolyticus* clearance rate was rapid during the first 24–48 hours of depuration. The non-pathogenic *V. parahaemolyticus* strain combination was reduced at a comparable or slower rate than the pathogenic combination, supporting its suitability as a surrogate for commercial validations. These findings support depuration as an

**Data availability statement:** All data for inoculation and depuration trials are available at figshare.com: https://doi.org/10.6084/m9.figshare.28710848.v1.

**Funding:** Taylor Shellfish Farms (Seattle, WA; https://www.taylorshellfishfarms.com) provided funding and the pilot-scale depuration system used in this study, as well as provided input on depuration parameters to evaluate. Additional funding provided by a United States Department of Agriculture National Institute of Food and Agriculture grant (2019-67017-29589; https://www.nifa.usda.gov) awarded to CBS supported these research efforts. Funders had no role in data collection and analysis, decision to publish, or preparation of the manuscript.

**Competing interests:** The authors have declared that no competing interests exist.

effective *V. parahaemolyticus* mitigation strategy in live oysters; though optimization of parameters, including temperature and duration, is needed to meet reduction targets in all species.

## Introduction

*Vibrio parahaemolyticus* is a foodborne pathogen found globally in marine environments [1]. This bacterium is a major cause of gastroenteritis associated with consuming raw or undercooked seafood, particularly raw oysters [2,3]. The first reported seafood-related outbreak of *V. parahaemolyticus* occurred in Japan in 1950 [4]. In subsequent decades, numerous sporadic outbreaks on nearly all continents have occurred [5]. Incidences of reported *Vibrio* spp. infections in the United States rose from the mid-1990s through 2010, with an increase in *V. parahaemolyticus* infections driving this increase [2,6].

*V. parahaemolyticus* favors growth in marine waters warmer than 15 °C. As sea temperatures increase, such conditions will be more common in regions traditionally considered too cool to support significant levels of this species [7]. Indeed, a *V. parahaemolyticus* outbreak was reported for the first time in Alaska in 2004, and studies have indicated that *Vibrio* populations in certain coastal waters, including the North Sea, are on the rise [3,8,9]. As conditions favoring *Vibrio* spp. continue to increase in coastal waters, the potential risk of human infections could also rise. To reduce the risk of foodborne *V. parahaemolyticus* infection, mitigation measures are needed to treat seafood, particularly products that are likely to be consumed without cooking (e.g., raw oysters).

The US Food and Drug Administration (FDA) and the Interstate Shellfish Sanitation Conference (ISSC) regulate the harvesting and shipping of raw shellfish through the National Shellfish Sanitation Program (NSSP). This federal and state cooperative program seeks to improve the safety of shellfish destined for human consumption and provides standards for raw or post-harvest processed (PHP) shellfish. PHP is recommended by the FDA to reduce the likelihood of *Vibrio*-associated illnesses linked to raw shellfish [10]. For companies to attach labeling claims to shellfish treated with PHP, they must demonstrate that the PHP method used to treat shellfish consistently meets NSSP guidelines. For oysters sold in the United States, these standards mandate a reduction of at least 3.52 log MPN/g in *V. parahaemolyticus* levels and a final concentration in treated oysters of < 30 MPN/g. Additionally, PHP can allow for the sale of oysters originating from sites closed for harvest due to the connection to reports of *V. parahaemolyticus*-associated illness; a > 3.0-log reduction is required to meet this standard for Pacific Coast oysters. A validation study plan approved by the ISSC is necessary to demonstrate the efficacy of the proposed PHP method [10].

Current FDA-approved PHP methods include freezing and frozen storage, heat treatment, and high hydrostatic pressure [11,12]. While these methods effectively

reduce *Vibrio* levels, they also kill the oysters. Consequently, the organoleptic properties of the oysters change, which may reduce their desirability to consumers [13].

In contrast, depuration, the process of holding live shellfish in sterile seawater to allow the shellfish to purge bacteria over time, does not kill oysters and thus may preserve qualities desired by consumers. Depuration systems usually use ultraviolet (UV) light, chlorine, or ozone to sterilize seawater, although not in all cases [14–16]. For decades, ambient-temperature depuration has been used to reduce bacteria levels in oysters, particularly pollution-associated enteric bacteria such as *Escherichia coli* [17]. However, multiple studies have indicated the inefficacy of ambient-temperature depuration in reducing *V. parahaemolyticus* [15,18]. As *V. parahaemolyticus* thrives at temperatures greater than 15 °C, much research has focused on lower-temperature depuration treatments to reduce this bacterium in an environment not conducive to its growth [19,20].

Studies of high-salinity and low-temperature depuration of oysters infected with *V. parahaemolyticus* have suggested promising results, with some treatments and trials successfully achieving the PHP standards set by the NSSP [21–23]. However, results are often inconsistent, and studies primarily use oysters infected with known clinical isolates [19,20,24]. Moreover, studies investigating the depuration of environmental *V. parahaemolyticus* strains have indicated that reducing these strains may be more challenging than reducing clinical strains, as environmental strains may be better adapted to survival within oyster tissues [25]. Additionally, depuration rates of *V. parahaemolyticus* in oysters may vary by species or subspecies, as species of oysters present differ in their physiology, activity, and response to temperature [26,27]. In a 2022 review of depuration parameters for reducing *V. parahaemolyticus*, the conditions found to be most effective were water temperatures < 20 °C, salinity from 25–32.2 ppt, depuration duration of four to six days, and non-static seawater conditions [28]. In the present study, we evaluated efficacy of depuration in a recirculating depuration system at temperatures from 5–13 °C, salinity of 35 ppt, and trial durations of five to seven days.

The objectives of this study were to first develop a consistent inoculation method that could achieve *V. parahaemolyticus* concentrations in oysters sufficient to support subsequent validation of a commercial depuration system (> 5 log MPN/g). This was assessed for three species of oysters: *Crassostrea gigas, C. sikamea,* and *C. virginica*. Additionally, the stability of the *V. parahaemolyticus* inoculum in infected oysters under simulated refrigerated transport conditions was evaluated. Secondly, this study sought to demonstrate the suitability of non-pathogenic strains of *V. parahaemolyticus*, originally isolated from oysters, as a surrogate for pathogenic strains for the eventual validation of commercial depuration systems. This was achieved by comparing the relative efficacy of depuration to reduce non-pathogenic and pathogenic strains in the three species of oysters listed above.

## Methods

### *V. parahaemolyticus* strain selection and growth rate comparisons

Pathogenic and non-pathogenic strains of *V. parahaemolyticus* were provided by Oregon State University's Astoria Seafood Laboratory and the Washington State Department of Health (Table 1). The growth rates of each strain were determined before conducting inoculation trials. All strains were revived from −80°C stock cultures of Tryptic Soy Broth + 1.5% NaCl (TSB-salt; Neogen, Lansing, MI) supplemented with 35% glycerol by transferring a loopful of culture to 10 mL TSB-salt and incubating at 35°C for 18–24 h. Then, one loopful of culture was streaked onto thiosulfate-citrate-bile salts-sucrose (TCBS; Neogen) and incubated at 35°C for 24 h. A single colony was picked and transferred to TSB-salt and incubated at 35°C for 4 h to achieve a cell density of 8–9 log CFU/mL. Aliquots of cultures (1 mL x 3) were centrifuged at 10,000 *x g* for 1 min, the supernatant discarded, and each culture resuspended in alkaline peptone water (APW, pH 8.5), sterile seawater (autoclaved natural seawater), or TSB-salt. Cultures were diluted to 3–4 log CFU/mL in the same medium and 100 µL of each diluted strain in each medium was transferred to a sterile 96-well plate in triplicate. The plate was placed into a microplate reader (FilterMax F5, Molecular Devices, San Jose, CA) set to 35°C and optical density (OD) was measured at 595 nm every 5 min for 24 h. Resulting data was used to evaluate potential differences in growth rates for cocktail preparation for oyster inoculation studies.

**Table 1. Bacterial strain, source information and virulence characteristics of *Vibrio parahaemolyticus* used in this study.**

| Inoculation Cocktail | Strain | Virulence Factor | Original Isolation Source |
|---|---|---|---|
| Pathogenic | 10290 | *tdh+, trh+* | Clinical (WA) |
| | 10292 | *tdh+, trh +* | Clinical (WA) |
| | 10293 | *tdh+, trh +* | Clinical (WA) |
| | BE 98–2029 | *tdh+, trh -* | Clinical (TX) |
| | O27-1c1 | *tdh+, trh +* | Clinical (OR) |
| Non-pathogenic | S13-052 | *tdh-, trh-* | Pacific Oyster (North North Bay, WA) |
| | S13-059 | *tdh-, trh-* | Pacific Oyster (Montanos Beach, Hood Canal #7, WA) |
| | S13-060 | *tdh-, trh-* | Pacific Oyster (Union, Annas Bay, WA) |
| | S13-075 | *tdh-, trh-* | Pacific Oyster (Chapman Cove, Oakland Bay, WA) |
| | S13-078 | *tdh-, trh-* | Pacific Oyster (Samish Bay – East, WA) |

## Oyster inoculation cocktail preparations

Individual strains of *V. parahaemolyticus* were revived from frozen storage (−80°C) by transferring approximately 10 µL into 10 mL of TSB-salt and incubating at 35°C for approximately 18 h. These cultures were then streaked on TCBS agar and incubated at 35°C for 24 h. Single colonies of each strain were selected, transferred to TSB-salt (20 mL), and incubated at 35°C for 12 h. Cultures were centrifuged at 3900 x $g$ for 5 minutes, the supernatant discarded, and the pellet resuspended in 20 mL sterile seawater (ASW). The optical density ($OD_{600nm}$) of each resuspended culture was measured using a spectrophotometer (Genesys 50, Thermo Fisher Scientific, Waltham, MA). Individual strains representing low, middle, and high $OD_{600}$ values were enumerated using standard serial dilution and spread plating on TSB-salt with incubation at 35°C for 24 h to correlate $OD_{600nm}$ with cell density to standardize cell densities. For cocktail preparation, each resuspended culture was adjusted to an $OD_{600nm}$ between 1.0 and 1.5 (8.0–8.5 log CFU/mL) by diluting with additional sterile seawater. Equal volumes of the five pathogenic or non-pathogenic strains were combined and mixed thoroughly to create inoculation cocktails. The total cell densities of each mixture were confirmed by serial dilution and spread plating. For each inoculation and depuration trial, bacterial cultures were initiated from frozen stock cultures.

## Oyster preparation, inoculation, and inoculum stability

Oysters of three species (*C. gigas, C. sikamea, C. virginica*) harvested from commercial oyster beds in Washington state were cooled and shipped overnight to Oregon State University with the appropriate Oregon Department of Fish and Wildlife import permits. Upon arrival, oysters were rinsed with tap water and allowed to acclimate in aerated artificial seawater at ambient temperature (20–22°C) for approximately 24 h. During that time, oysters were monitored for gaping activity to demonstrate they remained viable.

Oysters of each species (n = 12) were randomly selected from each batch to assess for natural contamination with *V. parahaemolyticus*. Oysters were shucked with a sanitized oyster knife; the shell contents were combined in a sterile glass jar and the mass was recorded. Alkaline phosphate-buffered saline (APBS; pH 7.4) was added to the container at twice the volume of the oyster tissue. The contents of the jar was homogenized with an immersion blender for approximately 30 s. The homogenate was then transferred to a 1.6-L filter bag (Whirl-Pak, Madison, WI) and an aliquot (100 µL) of the homogenate was plated on TCBS agar. Plates were evaluated after 24 h of incubation at 35°C. Representative colonies with typical and atypical morphology were selected for identification by 16S rRNA Sanger sequencing.

Round (9 cm diameter; 13 cm height) or square (14 cm width; 8 cm height) plastic containers were lined with plastic bags and filled with 300 mL of autoclaved seawater. For the first inoculation experiment, one laboratory (Waite-Cusic laboratory) used autoclaved artificial seawater (Instant Ocean, Blacksburg, VA), while the other (Schubiger laboratory) used

autoclaved natural seawater. Subsequent inoculations for depuration trials were conducted in autoclaved natural seawater. Two to three mL of the previously described pathogenic or non-pathogenic combination of *V. parahaemolyticus* strains was added to each bag containing autoclaved seawater and mixed to achieve an approximate cell density of 6 log CFU/mL. Individual oysters were placed in each container and held at ambient temperature (20–22 °C) for 24 hours to allow *V. parahaemolyticus* to accumulate naturally in the oysters during filter-feeding. At a minimum, five oysters per species were inoculated with each *V. parahaemolyticus* cocktail. Additional *C. gigas* oysters were inoculated with the non-pathogenic cocktail and then stored at 4°C for up to seven days to monitor the stability of the inoculum.

To facilitate larger experiments, a batch inoculation procedure was compared with the individual oyster inoculation procedure using the non-pathogenic *V. parahaemolyticus* cocktail. For the batch inoculation, oysters (n = 40/species) were distributed into 27-L (58 cm length; 41 cm width; 15 cm height) plastic containers. Each container was filled with 12 L of autoclaved seawater, after which 120 mL of the non-pathogenic strain combination was added to achieve an approximate cell density of 6–7 log CFU/mL. Oysters were allowed to accumulate *V. parahaemolyticus* under the same conditions as described in this section. The initial inoculation density was assessed after 24 h of inoculation.

## Depuration treatment of oysters

A pilot-scale recirculating depuration system equipped with an 80-W UV-sterilizer (Pentair Aquatic Eco-Systems, Apopka, FL), 1-µm bag filters, a protein skimmer (AquaC, San Diego, CA), a water chiller (Arctica DA-2000B, JBJ Aquariums, Inglewood, CA), and a temperature regulator was used to assess the effect of depuration on *V. parahaemolyticus* clearance from inoculated oysters (Fig 1). The depuration system was filled with approximately 450 L of 1 µm-filtered seawater (salinity: 35 ppt) 24 h prior to the addition of inoculated oysters; the system was allowed to run during this period to ensure chilling and UV-sterilization of the seawater. Depuration trials were conducted at 5, 11, or 13 °C (Table 2). Trial temperatures were chosen to assess the efficacy of depuration at a range of temperatures below 15 °C, as *V. parahaemolyticus* is known to proliferate above that temperature threshold [3,7,29]. A constant flow rate of 95 L/min was used for all trials.

To assess the efficacy of depuration, inoculated oysters were evenly stocked into the system's four holding chambers. Two chambers held oysters inoculated with pathogenic *V. parahaemolyticus*, and the other two chambers held oysters inoculated with non-pathogenic *V. parahaemolyticus* (Fig 1). For each oyster species and *V. parahaemolyticus* treatment, 25 oysters were added for trials lasting five days, and 35 oysters for seven-day trials. After placing oysters in the holding chambers, the depuration system was covered with a plastic tarp to reduce possible aerosolization of *V.*

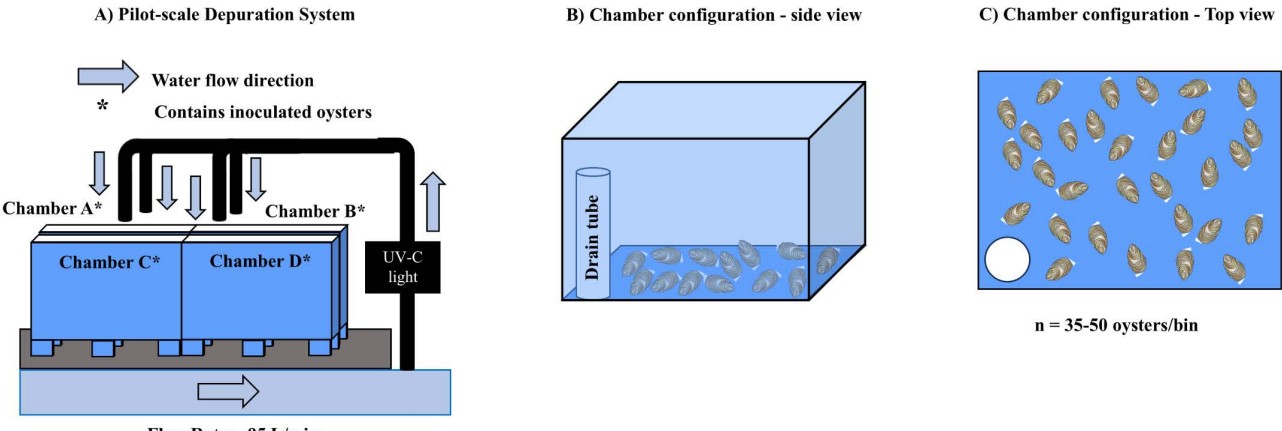

**Fig 1. Pilot-scale depuration trial configuration.** A) Overall system configuration, with seawater flowing from left (through filter and chiller unit, not shown). B) Side view of holding chamber, showing seawater drain tube. C) Overhead view of holding chamber, showing random distribution of oysters.

**Table 2. Depuration trial conditions.**

| Trial Number | Depuration Temperature (°C) | Depuration Length (d) | *Vibrio parahaemolyticus* cocktail[a] | Oyster Inoculation Method |
|---|---|---|---|---|
| 1 | 5 | 5 | NP, P | Individual |
| 2 | 11 | 5 | NP, P | Individual |
| 3 | 13 | 7 | NP, P | Individual |
| 4 | 13 | 7 | NP | Batch |

[a]P: Pathogenic cocktail of five clinical *V. parahaemolyticus* strains; NP: Non-pathogenic cocktail of five environmental *V. parahaemolyticus* strains.

*parahaemolyticus*. Every 24 h, five inoculated oysters of each of the three species and both *V. parahaemolyticus* combinations (30 oysters total) were collected from the system for bacterial enumeration. Sampling was conducted daily until the conclusion of the trial after five or seven days. Oysters were monitored daily during sample collection; oysters that were gaping and did not respond to touch were considered dead and were immediately removed from the system and destroyed.

## Enumeration of *V. parahaemolyticus*

*V. parahaemolyticus* cell densities were determined using serial dilution in phosphate buffered saline (PBS) with spread plating on TCBS agar for individual bacterial strains, strain mixtures, inoculated seawater, and inoculated and environmental control oysters. Plates were enumerated following incubation at 35°C for 24 h.

Individual oysters were shucked with a sanitized oyster knife. The entire shell contents, consisting of oyster tissue and liquor, were transferred to a large sterile container and the mass was recorded. PBS was added to the container at twice the volume of the oyster tissue, assuming the tissue's density was equivalent to 1 g/mL. The tissue and PBS were homogenized with a sanitized immersion blender for approximately 30 sec and then transferred to a 1.6-L filter bag (Whirl-Pak). Homogenate from the filter bag was further diluted and spread-plated as described in the previous paragraph.

## Identification of bacterial isolates from oysters

Representative colonies considered typical or atypical for *V. parahaemolyticus* were picked from TCBS plates from all environmental control oyster samples to determine natural *Vibrio* contamination levels. Atypical colonies with either abnormal in color or morphology were also selected from inoculated TCBS plates. Chosen colonies were streaked for isolation on TCBS agar plates and incubated at 35°C for up to 24 h. Single colonies were then transferred to 5 mL TSB-salt and incubated at 35°C for up to 24 h. DNA extractions were performed for each culture by aliquoting 750 µL into microcentrifuge tubes and centrifuging at 10,000 x g for 2 minutes. The supernatants were decanted, and each isolate was resuspended in 750 µL of nuclease-free water. The suspension was then boiled for 10 min in a heat block (Benchmark, Edison, NJ), followed by centrifugation at 10,000 x g for 2 min. The resulting supernatant (crude DNA extract) was stored at −20°C.

Sanger Sequencing of the 16S rRNA gene was performed for each isolate using 27F/1492R universal primers [30]. PCR was conducted in 25 µL reactions using Platinum Hot-Start Master Mix (Invitrogen, Carlsbad, CA). PCR conditions consisted of an initial denaturation at 95 °C for 5 min, followed by 30 cycles of 95°C for 30 sec, 51°C for 30 sec, and 72°C for 2 min, and a final extension at 72°C for 10 min. PCR products were purified with the DNA Fragment Extraction Kit (IBI Scientific, Dubuque, IA). Cleaned amplicons were submitted to Oregon State University's Center for Quantitative Life Sciences (CQLS, Corvallis, OR) for Sanger sequencing using both 27F and 1492R primers on an ABI 3730 Capillary Sequencer (Applied Biosystems, Waltham, MA). Raw sequencing reads were processed using SeqTrace to create consensus sequences [31]. Bacteria were identified at >99% similarity and >90% completeness using the 16S-based ID function of EzBioCloud [32].

## Statistical analysis

*V. parahaemolyticus* cell densities were converted to log CFU/g oyster tissue and liquor prior to graphing and statistical analysis. Reduction curves were modeled in JMP Pro (Version 18, JMP, Cary, NC) using the mechanistic growth function to predict reduction rate and maximum reduction (asymptote) comparable to Shen et al. [23]. Comparisons of depuration rates between non-pathogenic and pathogenic *V. parahaemolyticus* cocktails were accomplished by fitting linear models to the depuration data for each species using R statistical software (Version 4.0.3, R Project for Statistical Computing). Extra-sum-of-squares F-tests were then used to assess if an interaction between days of depuration and the *V. parahaemolyticus* strain combination produced a better fit of the model to the data. Interaction plots were produced to provide a visual estimate of possible interactions. To compare batch vs. individual inoculation methods, an F-test of equality of variances was used to determine if variances in the data sets were equal between batch- and individually-inoculated oysters. Mean inoculation cell densities were then compared using Welch's two-sample t-test.

## Results

### *V. parahaemolyticus* strain growth rates

The growth behaviors of individual *V. parahaemolyticus* strains (non-pathogenic and pathogenic) were determined in various growth media at 35 °C (Fig 2). All strains grew to OD between 0.3–0.6 in APW and 0.7–1.2 in TSB-salt and did not exponentially grow in sterilized seawater. One of the pathogenic strains (10290) consistently displayed an extended lag phase in both liquid media types. Two non-pathogenic strains (S13-075A and S13-052F) were slower than the other strains to reach the stationary phase. All strains reached maximum OD in TSB-salt after approximately 12 hours of incubation at 35 °C without aeration.

### Variability of oyster inoculation procedure

Oysters were inoculated simultaneously in two separate laboratory spaces, with one laboratory (Waite-Cusic) using artificial seawater (one inoculation) and the other (Schubiger) using autoclaved natural seawater (Fig 3). Inoculations performed in artificial seawater often resulted in significantly lower *V. parahaemolyticus* cell densities compared with inoculations performed in natural seawater. When comparing across all inoculations, there was significantly lower *V. parahaemolyticus* uptake in *C. virginica* compared to *C. gigas* and *C. sikamea* (p < 0.005). Oyster uptake of *V. parahaemolytics* was comparable for the pathogenic and non-pathogenic cocktails (p = 0.86).

Individual oyster inoculation procedures using natural seawater achieved >5 log CFU/g in nearly all oysters; however, inoculations using artificial seawater often failed to achieve this target, particularly for *C. sikamea* and *C. virginica*. The stability of the *V. parahaemolyticus* non-pathogenic cocktail in *C. gigas* inoculated in artificial seawater with up to 7 d of refrigerated storage was evaluated (Fig 3 inlay). The non-pathogenic *V. parahaemolyticus* inoculum remained consistently above 5 log CFU/g in *C. gigas* during the storage period (S1 Fig), indicating that this inoculation method and oyster handling procedure is promising for future commercial-scale validation studies.

When individual inoculation was compared to batch inoculation to simplify the inoculation protocol (Fig 4), both methods resulted in a >5 log CFU/g of *V. parahaemolyticus* in oyster tissue. However, batch inoculation yielded significantly lower *V. parahaemolyticus* concentrations in *C. sikamea* and *C. virginica* compared to the individual oyster inoculation procedure.

### Identification of natural microbiota of oysters recovered on TCBS

Non-target bacteria with the ability to grow on TCBS were isolated and speciated based on 16S rRNA sequence (Table 3). A total of 28 isolates were sequenced. Control oysters contained low levels (< 2 log CFU/g) of bacteria that are capable of growth on TCBS, and all displayed atypical morphologies that were easily distinguishable from *V. parahaemolyticus.*

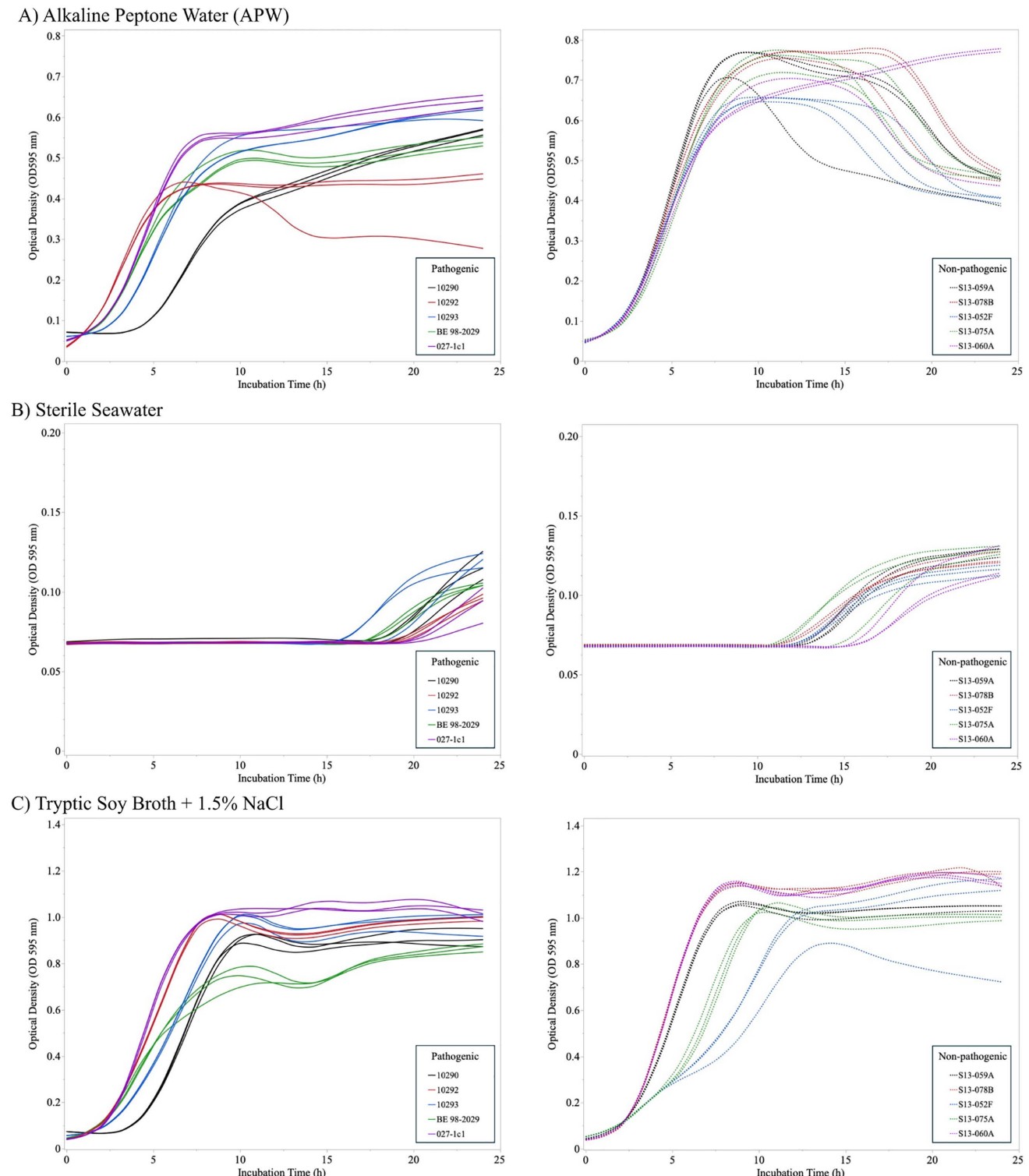

**Fig 2. Growth curves of individual pathogenic (left column) and non-pathogenic (right column)** *Vibrio parahaemolyticus* **strains in A) alkaline peptone water (APW), B) sterile seawater, and C) tryptic soy broth with 1.5% sodium chloride with incubation at 35°C for 24 h.** Lines represent growth in individual wells (n = 3/strain).

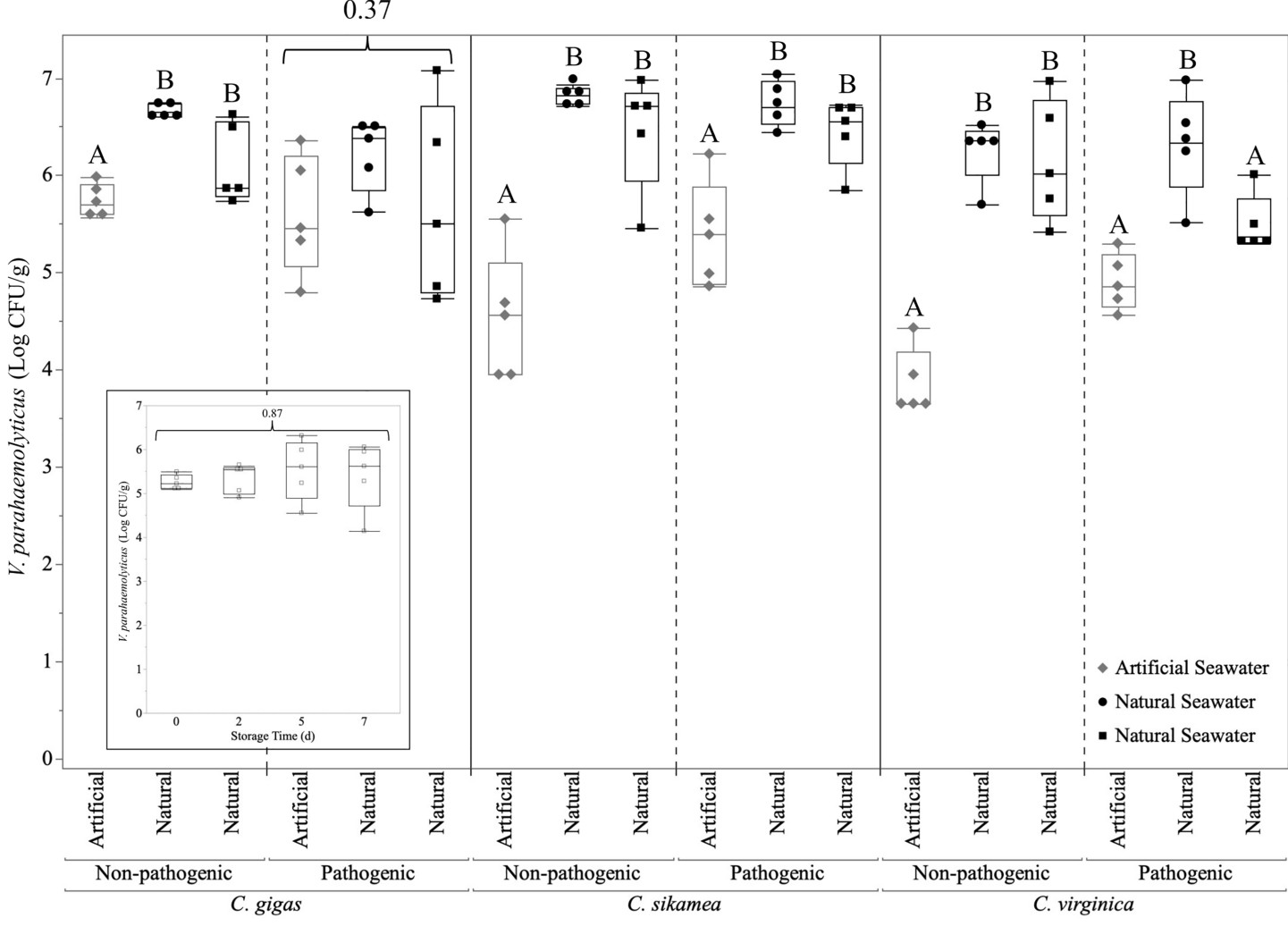

**Fig 3. Concentrations of *Vibrio parahaemolyticus* in oyster tissue (n = 5 oysters/species/cocktail/replicate) after 24 h of holding in artificial or natural seawater inoculated with non-pathogenic or pathogenic *V. parahaemolyticus* cocktails (6-7 log CFU/mL).** Data are shown from three biologically independent inoculations with artificial seawater inoculations in Waite-Cusic laboratory and autoclaved natural seawater inoculations in Schubiger laboratory. Statistical differences between inoculation procedures (One-way ANOVA with post-hoc Tukey's HSD; p < 0.05) are represented by different capital letters within each oyster species and *V. parahaemolyticus* cocktail. P-values are displayed for treatments that were not considered to be statistically significant. The graphical inlay shows the stability of non-pathogenic *V. parahaemolyticus* cell density in *C. gigas* oysters stored at 4°C for up to 7 d.

## Pilot-scale depuration trial results

Four depuration trials were conducted with the goal of identifying temperature and time conditions that would result in sufficient *V. parahaemolyticus* clearance to significantly reduce illness risk (e.g., > 3.0- or >3.52-log reduction). The results from the first depuration trial (5°C, 5 d) are shown in Fig 5. These conditions resulted in rapid clearance for the first 48 h; however, further depuration time did not result in additional reduction of *V. parahaemolyticus* in either *C. gigas* or *C. sikamea* oysters. These depuration conditions failed to predict sufficient *V. parahaemolyticus* reductions (>3.0- or 3.52-log reduction) that would support the validation of this process (Table 4). Maximum clearance rates and maximum reductions were comparable between pathogenic and non-pathogenic *V. parahaemolyticus* during Trial 1.

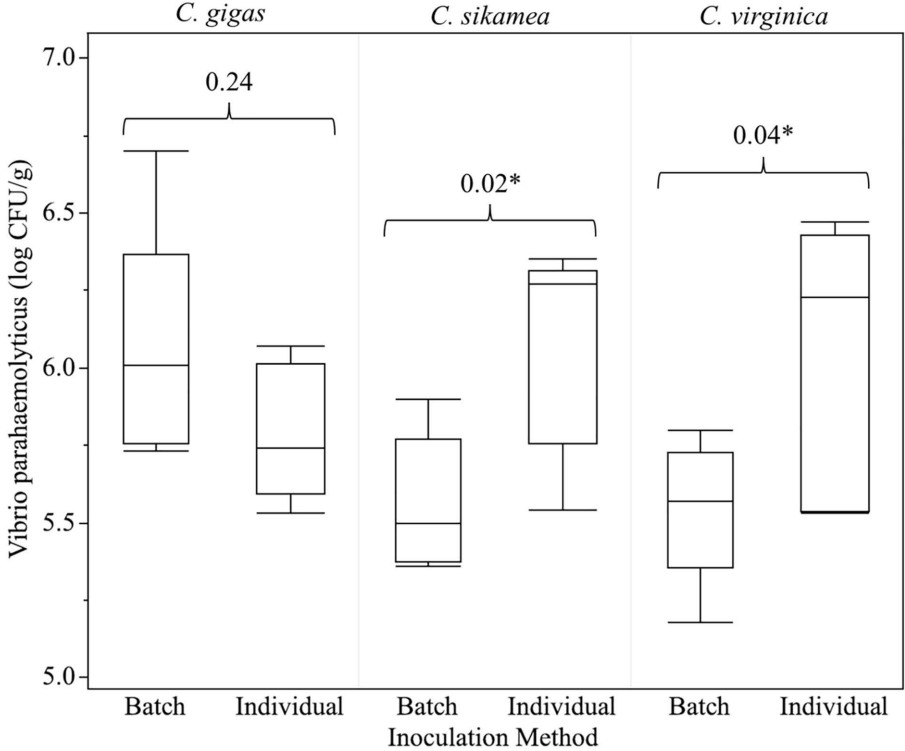

**Fig 4. Comparison of oyster inoculation procedures conducted on individual or batches of oysters using the non-pathogenic *Vibrio parahae-molyticus* cocktail.** Both inoculation methods achieved the minimum inoculation threshold for validation studies (5 log CFU/g) for all oyster species. Brackets indicate statistical comparisons between batch and individual oyster inoculation procedures by oyster species; numbers reported are *p*-values from two-sample t-tests. * indicates a significant difference (*p* < 0.05) between inoculation methods.

We hypothesized that oyster activity would be limited at 5°C and this dormancy could result in a lack of filtering and poor clearance of *V. parahaemolyticus.* A second depuration trial was conducted at a higher temperature (11°C, 5 d) with the results shown in Fig 6. At this slightly elevated temperature, maximum reduction rates were slightly slower; however, clearance continued for a longer period of time (72 h) and the maximum reductions of *V. parahaemolyticus* in Trial 2 were higher than in Trial 1 (Table 4). The 3.0-log reduction target was achieved for pathogenic *V. parahaemolyticus* in both *C. gigas* and *C. sikamea* and for non-pathogenic *V. parahaemolyticus* in *C. gigas.* The 3.52-log reduction target was only achieved for the pathogenic cocktail in *C. sikamea.* In all three oyster species, oyster clearance of the pathogenic and non-pathogenic was comparable or the non-pathogenic cocktail provided a conservative estimate of pathogenic clearance.

As the increased temperature (11°C) showed promising results, a third trial was conducted at a slightly elevated temperature (13°C) for a longer duration (7 d) with the results shown in Fig 7. *V. parahaemolyticus* clearances were comparable at 13°C and 11°C for both maximum reduction rates and maximum total reductions (Table 4). At 13°C, all three oyster species tested achieved the 3.0-log reduction target of the pathogenic cocktail. Significant clearance (>3.0- and >3.52-log reduction) of non-pathogenic *V. parahaemolyticus* was achieved by *C. sikamea* under these depuration conditions. However, this level of clearance efficacy was only achieved for this oyster species. It is important to note there was an increase in mortality of *C. gigas* during this trial with none of the oysters retaining viability after 6 d of depuration. Mortality of *C. gigas* was not observed in previous depuration trials, and mortality of the other oyster species in this trial was limited

**Table 3. Non-*Vibrio parahaemolyticus* bacterial species recovered on Thiosulfate-citrate-bile salts-sucrose (TCBS) from oysters received from commercial grow beds in Washington State. Isolates were speciated by Sanger sequencing of the 16S rRNA gene.**

| Oyster Species | Bacterial Species |
| --- | --- |
| *Crassostrea gigas* | *Micrococcus luteus* |
| | *Vibrio alginolyticus* |
| | *Vibrio anguillarum* |
| | *Vibrio azureus* |
| | *Vibrio hyugaensis* |
| | *Vibrio pacinii* |
| *Crassostrea sikamea* | *Vibrio anguillarum* |
| | *Vibrio aestuarianus* |
| | *Vibrio neocaledonicus* |
| *Crassostrea virginica* | *Oceanisphaera marina* |
| | *Vibrio alginolyticus* |
| | *Vibrio anguillarum* |
| | *Vibrio neocaledonicus* |

to one *C. virginica* individual from the non-pathogenic *V. parahaemolyticus* treatment. It is possible that the high mortality of *C. gigas* in this trial was due to environmental conditions such as temperature and water quality at the time and location of harvest, in addition to stress from shipping to the depuration site.

Due to the promising results from this trial, these depuration conditions were repeated using oysters inoculated with the non-pathogenic *V. parahaemolyticus* cocktail (Trial 4; Fig 8). Oyster clearance of *V. parahaemolyticus* in Trial 4 (13°C, 7 d) was more similar to the first trial (Trial 1; 5°C, 5 days) than to Trial 3 (13°C, 7 d). *V. parahaemolyticus* reduction rates were initially quite fast but effectively halted after 48 h of depuration. The lower 3-log reduction target of *V. parahaemolyticus* was not achieved for any of the oyster species tested (Table 4).

## Discussion

Results of the pilot-scale depuration trials indicate that recirculating depuration can significantly reduce *V. parahaemolyticus* levels in live oysters and have the potential to reach PHP labeling claim guidelines established under the NSSP [10]. In particular, the results of the second depuration trial also suggest that depuration could prove suitable for the treatment of oysters harvested from areas closed due to *V. parahaemolyticus* illnesses, which requires a reduction of > 3.0 log CFU/g. However, depuration did not achieve the targeted > 3.52 log CFU/g reduction of *V. parahaemolyticus* and reduction below the detection limit of 30 MPN/g that would be required for a PHP labeling claim on harvested oysters. Reduction rates varied between different species of oysters when treated under identical conditions, with depuration of *C. virginica* taking considerably longer than the depuration of *C. gigas* and *C. sikamea* and never reaching a > 3.0 log CFU/g reduction in any trial. Previous depuration research aiming at reducing *V. parahaemolyticus* in *C. virginica* achieved a reduction of 2.1 log MPN/g after 48 hours at 15 °C [24]. This aligns with the results of our depuration trials with *C. virginica*, where we observed final asymptotic reductions of 1.99–2.82 log CFU/g and 2.92–3.07 log CFU/g reductions in non-pathogenic and pathogenic *V. parahaemolyticus,* respectively.

While significant reduction of *V. parahaemolyticus* in oysters is possible, consistent reductions below the detection limit did not occur in any treatment or trial; while some individual oysters reached the detection threshold, average concentrations remained > 30 CFU/g. Studies present in the literature have achieved a > 3.52 log CFU/g reduction and < 30 MPN/g final concentration in *C. gigas*, indicating the potential to reach this threshold with modifications to the methods used

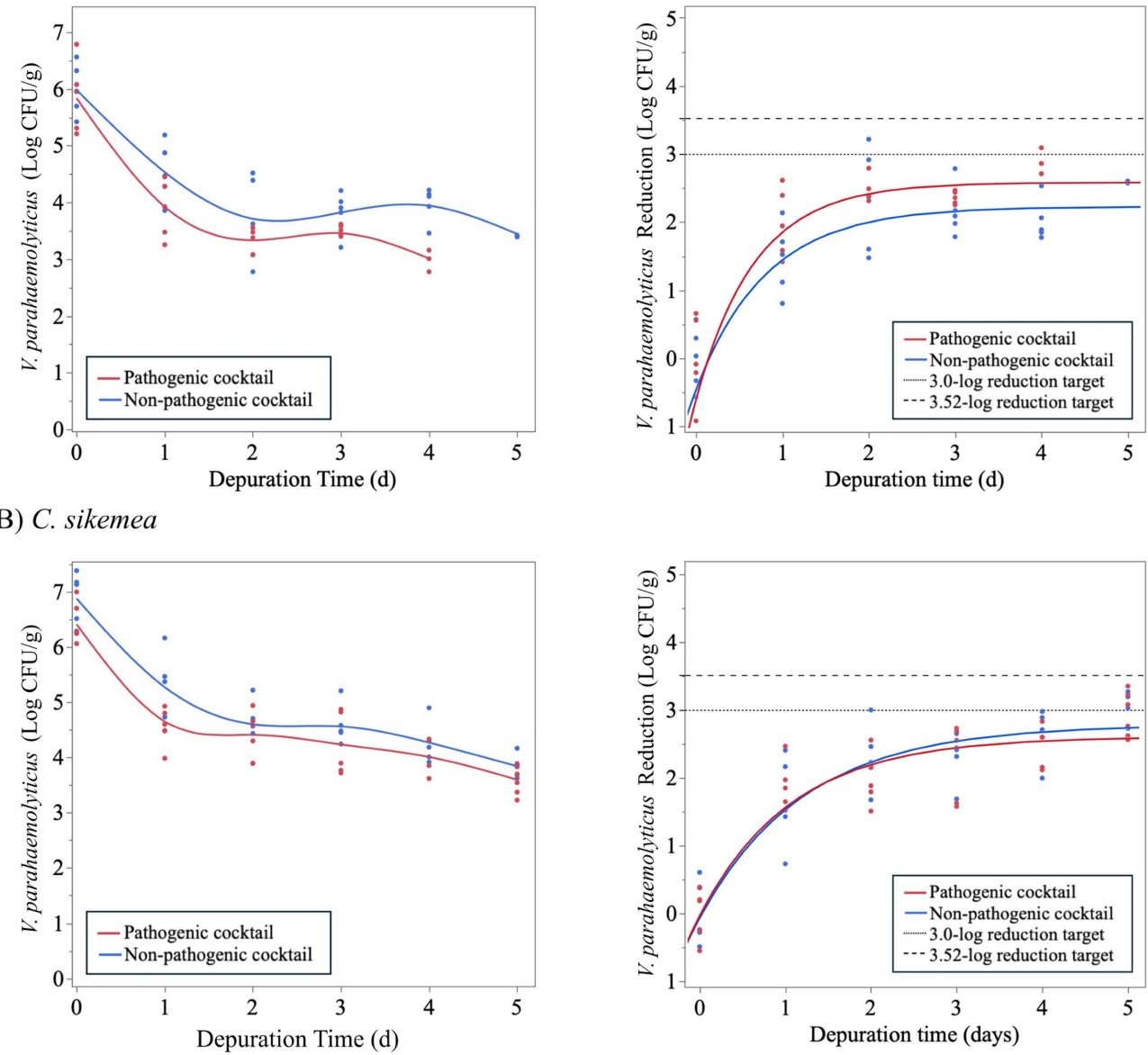

**Fig 5. Change in non-pathogenic and pathogenic *Vibrio parahaemolyticus* populations in (A) *Crassostrea gigas* and (B) *Crassostrea sikamea* during depuration at 5°C for up to 5 days (Trial 1).** Figures of the left indicate the *V. parahaemolyticus* survivors. Figures on the right indicate *V. parahaemolyticus* reduction with the colored lines indicating the fit model for the pathogenic and non-pathogenic *V. parahaemolyticus* clearance. Dashed lines indicate 3.0- and 3.52-log reduction targets for depuration validation.

[21,23]. Further investigation is needed under varying depuration parameters (salinity, temperature) to determine ideal conditions for reduction of *V. parahaemolyticus* in each oyster species for which PHP is desired.

The comparison of non-pathogenic and pathogenic *V. parahaemolyticus* depuration rates suggests that future studies could be conducted with the non-pathogenic *V. parahaemolyticus* strain combinations only, as the reduction of non-pathogenic strains was comparable or slower than pathogenic strains in all trials, with all oyster species tested. Most

**Table 4. *Vibrio parahaemolyticus* reduction estimates and predicted depuration times to achieve targeted microbial clearance goals for each depuration trial as modeled using the nonlinear mechanistic growth function with inverse prediction. NA = Not achieved; the model estimated a maximum reduction that was less than the desired target reduction.**

| Trial | Oyster Species | *V. parahaemolyticus* cocktail | Maximum Reduction Rate (Log CFU/g/d) (95% CI) | Maximum Reduction (Log CFU/g) (95% CI) | Predicted Depuration time to 3-log Reduction (d) (95% CI) | Predicted Depuration time to 3.52-log Reduction (d) (95% CI) |
|---|---|---|---|---|---|---|
| 1:<br>5°C<br>5 d | *C. gigas* | Pathogenic | 1.43 (0.57-2.28) | 2.62 (2.29-2.97) | NA | NA |
| | | Non-pathogenic | 1.20 (0.47-1.92) | 2.27 (1.96-2.58) | NA | NA |
| | *C. sikamea* | Pathogenic | 0.89 (0.45-1.33) | 2.65 (2.33-2.98) | NA | NA |
| | | Non-pathogenic | 0.79 (0.40-1.18) | 2.83 (2.44-3.22) | NA | NA |
| 2<br>11°C 5 d | *C. gigas* | Pathogenic | 0.68 (0.28-1.09) | 3.26 (2.70-3.82) | 3.6 (2.1-5.1) | NA |
| | | Non-pathogenic | 1.14 (0.52-1.76) | 3.05 (2.70-3.41) | 3.5 (−0.9-8.0) | NA |
| | *C. sikamea* | Pathogenic | 1.14 (0.61-1.68) | 3.81 (3.44-4.18) | 1.3 (0.9-1.8) | 2.2 (1.4-3.1) |
| | | Non-pathogenic | 1.30 (0.51-2.09) | 2.99 (2.65-3.34) | NA | NA |
| | *C. virginica* | Pathogenic | 0.91 (0.46-1.36) | 2.92 (2.56-3.28) | NA | NA |
| | | Non-pathogenic | 0.53 (0.18-0.88) | 2.83 (2.14-3.51) | NA | NA |
| 3<br>13°C<br>7 d | *C. gigas* | Pathogenic | 1.25 (0.69-1.80) | 3.31 (3.03-3.58) | 1.9 (1.2-2.6) | NA |
| | | Non-pathogenic | 0.65 (0.30-1.01) | 2.67 (2.25-3.09) | NA | NA |
| | *C. sikamea* | Pathogenic | 0.68 (0.45-0.90) | 3.37 (3.10-3.63) | 3.2 (3.6-3.9) | NA |
| | | Non-pathogenic | 0.97 (0.67-1.26) | 3.52 (3.31-3.73) | 2.0 (1.5-2.4) | 7.8 (−106-122) |
| | *C. virginica* | Pathogenic | 0.53 (0.26-0.80) | 3.07 (2.64-3.51) | 7.0 (−1.3-15.3) | NA |
| | | Non-pathogenic | 0.60 (0.16-1.03) | 1.99 (1.59-2.39) | NA | NA |
| 4<br>13°C<br>7 d | *C. gigas* | Non-pathogenic | 1.35 (0.57-2.13) | 2.43 (2.21-2.65) | NA | NA |
| | *C. sikamea* | | 1.97 (0.44-3.50) | 2.35 (2.15-2.55) | NA | NA |
| | *C. virginica* | | 0.93 (0.39-1.46) | 2.23 (1.98-2.48) | NA | NA |

published studies on depuration have focused on reducing pathogenic *V. parahaemolyticus* isolates or have used oysters that are naturally contaminated with *V. parahaemolyticus* [20–22,25]. In this study, we used environmental isolates to inoculate oysters, allowing a more direct comparison of pathogenic and non-pathogenic isolates in the same depuration system. Non-pathogenic *V. parahaemolyticus* was found to reduce at a comparable rate to pathogenic strains, allowing for future commercial-scale depuration trials to be conducted without the concern of introducing pathogenic strains of

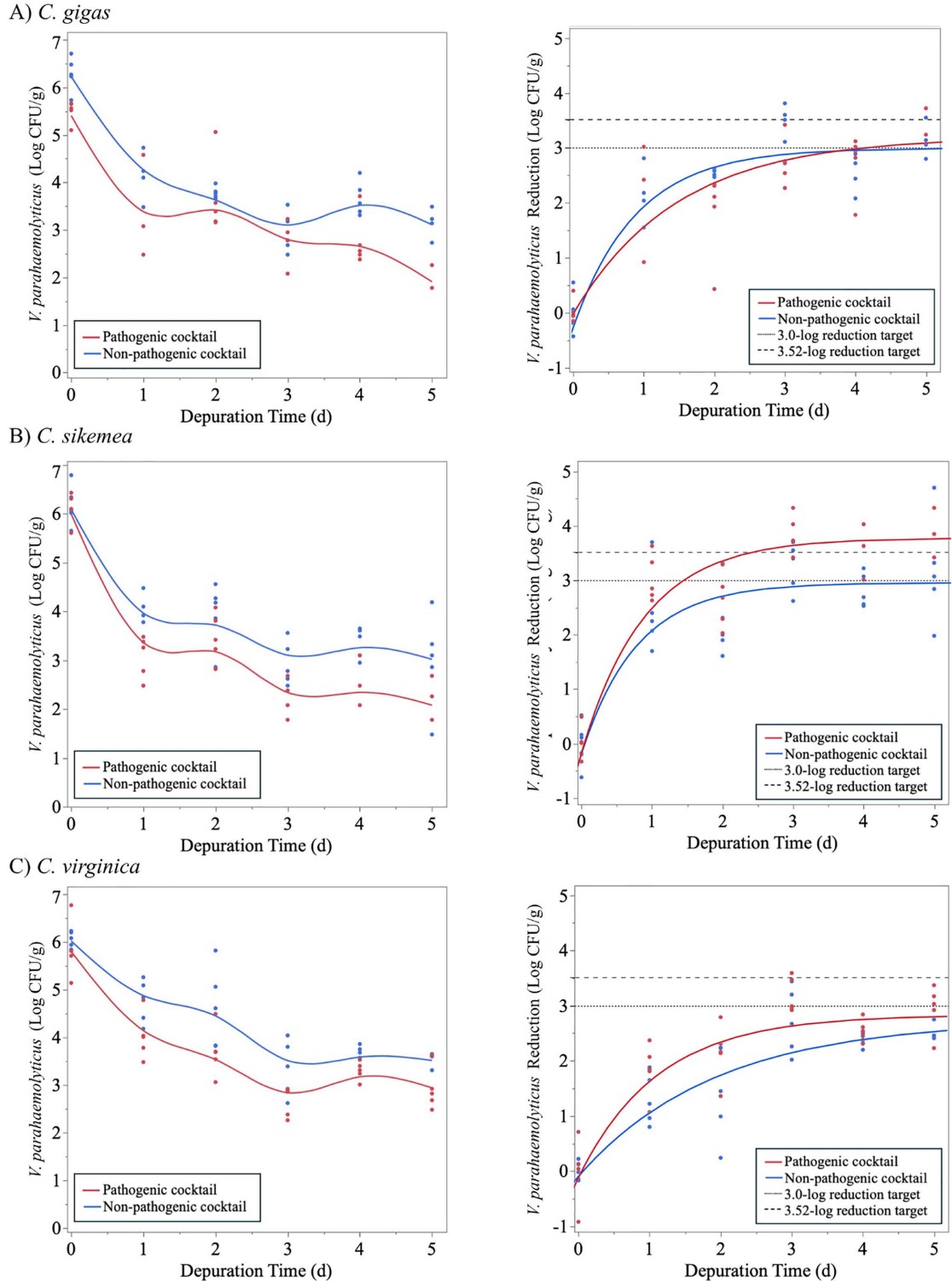

**Fig 6. Change in non-pathogenic and pathogenic *Vibrio parahaemolyticus* populations in (A) *Crassostrea gigas*, (B) *Crassostrea sikamea*, and (C) *Crassostrea virginica* during depuration at 11°C for up to 5 days (Trial 2).** Figures of the left indicate the *V. parahaemolyticus* survivors. Figures on the right indicate *V. parahaemolyticus* reduction with the colored lines indicating the fit model for the pathogenic and non-pathogenic *V. parahaemolyticus* clearance. Dashed lines indicate 3.0- and 3.52-log reduction targets for depuration validation.

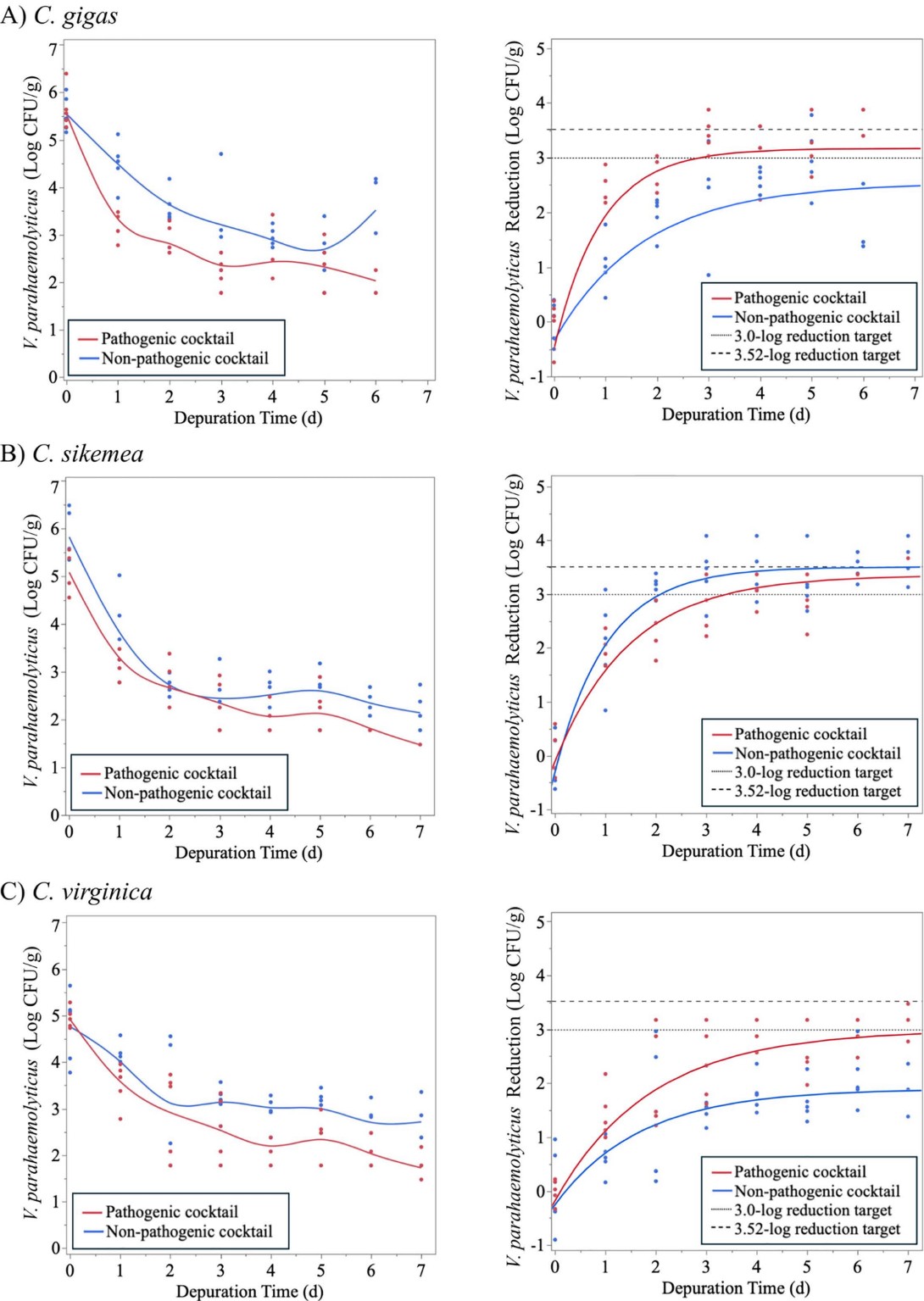

**Fig 7. Change in non-pathogenic and pathogenic *Vibrio parahaemolyticus* populations in (A) *Crassostrea gigas*, (B) *Crassostrea sikamea*, and (C) *Crassostrea virginica* during depuration at 13°C for up to 7 days (Trial 3).** Figures of the left indicate the *V. parahaemolyticus* survivors. Figures on the right indicate *V. parahaemolyticus* reduction with the colored lines indicating the fit model for the pathogenic and non-pathogenic *V. parahaemolyticus* clearance. Dashed lines indicate 3.0- and 3.52-log reduction targets for depuration validation.

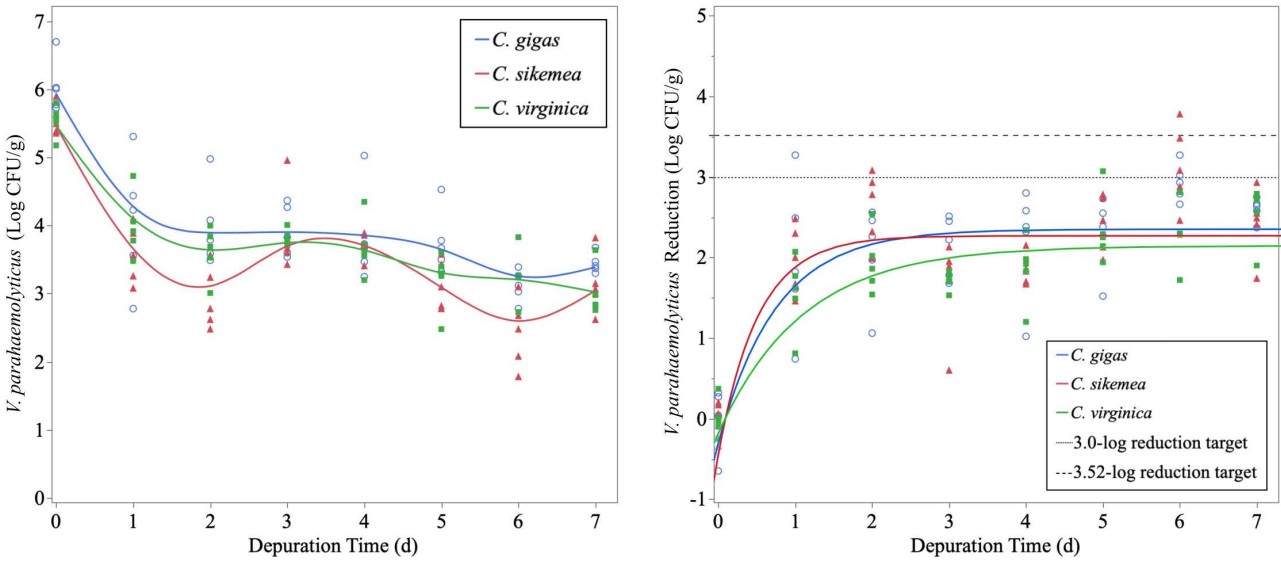

**Fig 8. Change in non-pathogenic *Vibrio parahaemolyticus* populations in *Crassostrea gigas, Crassostrea sikamea*, and *Crassostrea virginica* during depuration at 13°C for up to 7 days (Trial 4).** Inoculation was performed using the batch method. Figures of the left indicate the *V. parahaemolyticus* survivors. Figures on the right indicate *V. parahaemolyticus* reduction with the colored lines indicating the fit model for the pathogenic and non-pathogenic *V. parahaemolyticus* clearance. Dashed lines indicate 3.0- and 3.52-log reduction targets for depuration validation.

*V. parahaemolyticus* into facilities. This also allows for simplified procedures to safely dispose of effluent seawater from the system.

Findings from the refrigerated storage study demonstrated that *V. parahaemolyticus* concentrations in oyster tissue remain relatively constant during storage at 4 °C, supporting the suitability of refrigerated transport as a suitable method for shipping oysters to laboratories for commercial-scale validation studies. This is in accordance with previously published results, which found non-significant changes in *V. parahaemolyticus* cell densities after storage on ice [21].

The results of the initial inoculation study, and inoculation data collected from subsequent depuration trials, demonstrated that the procedure developed here was suitable for consistently achieving a > 5 log CFU/g *V. parahaemolyticus* concentration in oyster tissue when inoculation was conducted in natural autoclaved seawater. However, there were exceptions to this finding in some trials and oyster species, particularly the initial inoculation levels observed for *C. virginica* in depuration trial 3. This could have been due to this batch of oysters being in poorer overall health than other batches, as demonstrated by mortalities of one *C. virginica* and fourteen *C. gigas* individuals after six days of depuration. Additionally, the failure of *C. virginica* and *C. sikamea* to accumulate > 5 log CFU/g when inoculated in artificial seawater is notable, as previous studies succeeded in achieving *V. parahaemolyticus* cell densities of > 5 log CFU/g in artificial seawater [22–24]. In addition, inoculation for all depuration trials was conducted in autoclaved natural seawater and consistently achieved > 5 log CFU/g cell densities. Therefore, it is possible that slight differences in artificial seawater preparation, sea salt brands, or municipal water sources might be responsible for these discrepancies.

Analysis of environmental control oysters demonstrated that there was no confounding of results obtained in inoculation and depuration trials by already-present *V. parahaemolyticus*. Very low levels of bacteria capable of growth on TCBS were present in controls, and colonies presented morphologies that were easy to distinguish from *V. parahaemolyticus.* The inoculation and depuration trials were conducted from April – June 2021; oysters harvested during different times of the year, particularly late summer, may contain higher levels of *Vibrio* species, including *V. parahaemolyticus,* than were observed in this study. Many outbreaks of *V. parahaemolyticus*-related illness occur in August or September, and

*V. parahaemolyticus* has been known to be found in higher concentrations in these months; therefore, oysters harvested during this time would be expected to have a greater background level of *Vibrio* spp. [2,29].

In conclusion, recirculating depuration systems have the potential to reduce *Vibrio parahaemolyticus* levels in live oysters to levels mandated for PHP by the NSSP [10]. However, depuration rates vary significantly between oyster species and individual oysters. Further work is necessary to identify optimal parameters for the depuration of oysters, particularly for each species of interest. This study also demonstrated that non-pathogenic strains of *V. parahaemolyticus* are suitable for future depuration studies, as these strains are comparable or even more challenging to depurate than pathogenic strains; this could allow for future depuration studies to be conducted without requiring the handling and use of pathogenic isolates.

## Supporting information

**S1 Fig. Changes in concentration of non-pathogenic *V. parahaemolyticus* in oysters during storage at 4°C for up to 7 days.** Day 0 samples were taken after oysters were held in contaminated water for 24 hrs.
(TIFF)

## Acknowledgments

The authors would like to thank members of the Waite-Cusic Food Safety and Quality Systems Laboratory (Corvallis, OR) for their assistance with oyster sample analysis. The authors thank Lynette Hawthorne (Oregon State University, Corvallis, OR) for administrative support and Tiffany Spendiff for assisting with *V. parahaemolyticus* enumeration. In addition, we thank Bill Dewey and Austin Docter (Taylor Shellfish, Shelton, WA), and Andy DePaola (Angelo DePaola Consulting, Coden, AL) for all the insightful discussions and ideas on which depuration parameters to test.

## Author contributions

**Conceptualization:** Spencer Lunda, Joy G. Waite-Cusic, Carla B. Schubiger.

**Data curation:** Spencer Lunda, Samantha Kilgore, Jennifer M. Hesser, Joy G. Waite-Cusic, Carla B. Schubiger.

**Formal analysis:** Spencer Lunda, Joy G. Waite-Cusic.

**Funding acquisition:** Joy G. Waite-Cusic, Carla B. Schubiger.

**Investigation:** Spencer Lunda, Samantha Kilgore, Jennifer M. Hesser, Joy G. Waite-Cusic, Carla B. Schubiger.

**Methodology:** Spencer Lunda, Joy G. Waite-Cusic, Carla B. Schubiger.

**Project administration:** Joy G. Waite-Cusic, Carla B. Schubiger.

**Resources:** Joy G. Waite-Cusic, Carla B. Schubiger.

**Supervision:** Joy G. Waite-Cusic, Carla B. Schubiger.

**Validation:** Spencer Lunda, Samantha Kilgore, Joy G. Waite-Cusic, Carla B. Schubiger.

**Visualization:** Spencer Lunda, Samantha Kilgore, Joy G. Waite-Cusic.

**Writing – original draft:** Spencer Lunda.

**Writing – review & editing:** Spencer Lunda, Samantha Kilgore, Jennifer M. Hesser, Joy G. Waite-Cusic, Carla B. Schubiger.

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
