## [Decision Letter · Decision Letter 0]

28 May 2025

Dear Dr. Lunda,

Thank you for submitting your manuscript to PLOS ONE. After careful consideration, we feel that it has merit but does not fully meet PLOS ONE’s publication criteria as it currently stands. Therefore, we invite you to submit a revised version of the manuscript that addresses the points raised during the review process.

Please make the changes or additions that are requested by myself and the other reviewers.

We look forward to receiving your revised manuscript.

Kind regards,

Brett Austin Froelich, Ph.D

Academic Editor

PLOS ONE

 [Taylor Shellfish Farms (Seattle, WA; https://www.taylorshellfishfarms.com) provided funding and the pilot-scale depuration system used in this study, as well as provided input on depuration parameters to evaluate.

Additional funding provided by a United States Department of Agriculture National Institute of Food and Agriculture grant (2019-67017-29589; https://www.nifa.usda.gov) awarded to CBS supported these research efforts.]. 

Additional Editor Comments:

L 199-201. It seems the natural and artificial sea water terms are switched here, as natural seawater should not have a company associated with it.

Reviewers' comments:

Reviewer's Responses to Questions

**Comments to the Author**

1. Is the manuscript technically sound, and do the data support the conclusions?

Reviewer #1: Yes

Reviewer #2: Yes

2. Has the statistical analysis been performed appropriately and rigorously?

Reviewer #1: Yes

Reviewer #2: Yes

3. Have the authors made all data underlying the findings in their manuscript fully available?

Reviewer #1: Yes

Reviewer #2: Yes

4. Is the manuscript presented in an intelligible fashion and written in standard English?

Reviewer #1: Yes

Reviewer #2: Yes

Reviewer #1: Thank you for your time and for your interesting research review. Such a topic is very necessary Vibrio research oyster depuration application efforts. I do, however, suggest a few improvements should be made (see below).

Overall

This paper offers a valuable contribution to the body of knowledge concerning Vibrio mitigation. It is well written and structured, and successfully compares the effects of depuration across oyster species, evaluates non-pathogenic Vibrio as surrogates, and optimizes pilot-scale depuration. The results are particularly helpful for informing commercial depuration protocols for efforts to comply with NSSP standards for Vibrio reduction in oysters.

However, the clarity of the figures detracts from the overall impact. Even when downloaded, I could not really view the figures. Please submit images with better resolution. Additionally, the rationale for selecting the specific temperatures of 5, 11, and 13°C for depuration trials is not clearly explained. Please include a brief explanation to help readers understand why these temperature parameters were selected.

To further enhance the text, it would be useful to include data from the 2022 Campbell et al. review, Depuration of live oysters to reduce Vibrio parahaemolyticus and Vibrio vulnificus: A review of ecology and processing parameters. This review could be referenced in lines 113-122 or in your discussion. The authors also mention:

“Although many oyster depuration studies demonstrate significant variability when detailing the reduction of Vibrio vulnificus (VV) and Vibrio parahaemolyticus (VP), evaluation of these findings showed the greatest reductions of VV and VP were when processing time was from 4 to 6 days, water temperature was less than 20◦C, water salinity ranged from 25 to 32.2 ppt, and the water was flowing (non-static systems).”

[Campbell, V. M., Chouljenko, A., & Hall, S. G. (2022). Depuration of live oysters to reduce Vibrio parahaemolyticus and Vibrio vulnificus: A review of ecology and processing parameters. Comprehensive Reviews in Food Science and Food Safety, 21, 3480–3506. https://doi.org/10.1111/1541-4337.12969]

Overall, this is an excellent study with practical applications that would benefit the oyster industry. Thank you.

Reviewer #2: This manuscript reports pilot studies to better define the conditions to be used during larger-commercial scale depuration studies focusing on V. parahaemolyticus. The strains used as inoculum (pathogenic versus non-pathogenic), the inoculation approach (individual versus batch exposure), the autoclaved seawater used (artificial versus natural), the impact of the oyster species on the results as well as temperatures were evaluated during the study. I think the manuscript is well written and the experimental approach sound. My main concern relates to the method used to measure Vp concentrations (plating on TCBS) and I suggest discussing the limits of this approach and how this approach may be underestimating Vp concentrations.

Ln 451,…: Does this mean that the only criterion for the Pacific Coast is that a reduction of >3.0 log is demonstrated? Is the < 30 MPN/g concentrations not used? Please clarify.

Ln 228: Please include information relative to the number of oysters per chamber per species.

Ln 253-256: Following NSSP, the method used to quantify the targeted Vibrio species during PHP validation is a MPN approach. Please justify the choice of a TCBS-based approach.

Ln 253, …: Enumeration on TCBS can estimate total Vibrio counts, so how did Vp counts and concentrations were measured? Based on colonies color? Please include additional information. It seems that some sequencing was conducted but it is not clear how many colonies were sequenced per plate. These are critical information to assess if the concentrations reported are accurate. Please clarify.

Ln 365,…: the Methods indicated that the natural Vp concentrations was assessed during this study (prior to inoculation), but these data are not presented. Please include these data.

Ln 424: Such mortality is not expected in such a short time, any hypothesis regarding the factors involved? One can question the results of this trial since the other species were in the same tank. Were mortalities also observed in the other two species?

**Do you want your identity to be public for this peer review?** For information about this choice, including consent withdrawal, please see our Privacy Policy

Reviewer #1: No

Reviewer #2: No

---

## [Author Response · Author response to Decision Letter 1]

19 Jul 2025

Additional Editor Comments:

L 199-201. It seems the natural and artificial sea water terms are switched here, as natural seawater should not have a company associated with it.

- Thank you for pointing this out, we have corrected this by adding the manufacturer following “…artificial seawater”.

Reviewer #1: Thank you for your time and for your interesting research review. Such a topic is very necessary Vibrio research oyster depuration application efforts. I do, however, suggest a few improvements should be made (see below).

Overall

This paper offers a valuable contribution to the body of knowledge concerning Vibrio mitigation. It is well written and structured, and successfully compares the effects of depuration across oyster species, evaluates non-pathogenic Vibrio as surrogates, and optimizes pilot-scale depuration. The results are particularly helpful for informing commercial depuration protocols for efforts to comply with NSSP standards for Vibrio reduction in oysters.

However, the clarity of the figures detracts from the overall impact. Even when downloaded, I could not really view the figures. Please submit images with better resolution. Additionally, the rationale for selecting the specific temperatures of 5, 11, and 13°C for depuration trials is not clearly explained. Please include a brief explanation to help readers understand why these temperature parameters were selected.

- Thank you for these comments, we have provided higher-resolution figures that should allow for easier interpretation of results. We have also added a short explanation of why we tested these temperatures (after Line 230):

“Trial temperatures were chosen to assess the efficacy of depuration at a range of temperatures below 15 °C, as V. parahaemolyticus is known to proliferate above that temperature threshold.”

To further enhance the text, it would be useful to include data from the 2022 Campbell et al. review, Depuration of live oysters to reduce Vibrio parahaemolyticus and Vibrio vulnificus: A review of ecology and processing parameters. This review could be referenced in lines 113-122 or in your discussion. The authors also mention:

“Although many oyster depuration studies demonstrate significant variability when detailing the reduction of Vibrio vulnificus (VV) and Vibrio parahaemolyticus (VP), evaluation of these findings showed the greatest reductions of VV and VP were when processing time was from 4 to 6 days, water temperature was less than 20 °C, water salinity ranged from 25 to 32.2 ppt, and the water was flowing (non-static systems).”

[Campbell, V. M., Chouljenko, A., & Hall, S. G. (2022). Depuration of live oysters to reduce Vibrio parahaemolyticus and Vibrio vulnificus: A review of ecology and processing parameters. Comprehensive Reviews in Food Science and Food Safety, 21, 3480–3506. https://doi.org/10.1111/1541-4337.12969]

Overall, this is an excellent study with practical applications that would benefit the oyster industry. Thank you.

- Thank you for your comments, and for providing an additional review reference for citation. We have included this reference in Lines 113-122 to provide context of depuration parameters that are generally considered to be most effective for achieving targeted reductions:

- “In a 2022 review of depuration parameters for reducing V. parahaemolyticus, the conditions found to be most effective were water temperatures < 20 °C, salinity from 25-32.2 ppt, depuration duration of four to six days, and non-static seawater conditions [28]. In the present study, we evaluated efficacy of depuration in a recirculating depuration system at temperatures from 5-13 °C, salinity of 35 ppt, and trial durations of five to seven days.”

Reviewer #2: This manuscript reports pilot studies to better define the conditions to be used during larger-commercial scale depuration studies focusing on V. parahaemolyticus. The strains used as inoculum (pathogenic versus non-pathogenic), the inoculation approach (individual versus batch exposure), the autoclaved seawater used (artificial versus natural), the impact of the oyster species on the results as well as temperatures were evaluated during the study. I think the manuscript is well written and the experimental approach sound. My main concern relates to the method used to measure Vp concentrations (plating on TCBS) and I suggest discussing the limits of this approach and how this approach may be underestimating Vp concentrations.

- Thank you for your comments and review; we have provided responses to your specific comments below.

Ln 451,…: Does this mean that the only criterion for the Pacific Coast is that a reduction of >3.0 log is demonstrated? Is the < 30 MPN/g concentrations not used? Please clarify.

- For harvest from areas closed due to reports of illness associated with V. parahaemolyticus, harvest may be allowed if depuration can achieve a > 3.0 log reduction, but oysters treated with this method would not have a labeling claim attached indicating that post-harvest processing methods have been used. For the labeling claim, the 3.52 log reduction and < 30 MPN/g is required. To clarify this point, we have added a sentence following the one that you highlighted:

- “However, depuration did not achieve the targeted > 3.52 log CFU/g reduction of V. parahaemolyticus and reduction below the detection limit of 30 MPN/g that would be required for a PHP labeling claim on harvested oysters.“

Ln 228: Please include information relative to the number of oysters per chamber per species.

- We have added a sentence to provide information on numbers of oysters used for, in line 243 in the original manuscript:

“For each oyster species and V. parahaemolyticus treatment, 25 oysters were added for trials lasting five days, and 35 oysters for seven-day trials.”

- In addition, to clarify about stocking we added “evenly” to the sentence: “To assess efficacy of depuration, inoculated oysters were evenly stocked into the system’s four holding chambers.”

Ln 253-256: Following NSSP, the method used to quantify the targeted Vibrio species during PHP validation is a MPN approach. Please justify the choice of a TCBS-based approach.

- Thank you for this comment; we are aware of the MPN approach included in the NSSP. We have performed side by side enumeration of inoculated and depurated oyster samples using both MPN and TCBS plate counts and results were similar for all samples between the two methods. Enumeration on TCBS consumes less resources (supplies, labor, and time); therefore, this method was selected for use in this study.

Ln 253, …: Enumeration on TCBS can estimate total Vibrio counts, so how did Vp counts and concentrations were measured? Based on colonies color? Please include additional information. It seems that some sequencing was conducted but it is not clear how many colonies were sequenced per plate. These are critical information to assess if the concentrations reported are accurate. Please clarify.

Background Vibrio populations in oysters were very low (<2 log CFU/g) and inoculation levels were very high (~6 log CFU/g); therefore, the risk of the natural Vibrio contamination influencing the enumeration was insignificant. Therefore, all colonies on TCBS from inoculated oysters displayed typical V. parahaemolyticus morphology and were enumerated. Uninoculated oyster samples of all species had low levels (<2 log CFU/g) of bacteria that were capable of growth on TCBS that displayed morphologies that were atypical (color, size) in comparison to V. parahaemolyticus. Due to this low level of background contamination, there were few colonies to assess. A total of 28 isolates were sequenced with their identities shown in Table 3.

- In line 366, we have added the sentence: “A total of 28 isolates were sequenced”.

Ln 365,…: the Methods indicated that the natural Vp concentrations was assessed during this study (prior to inoculation), but these data are not presented. Please include these data.

- Thank you for this comment; the primary purpose of enumerating uninoculated oysters was to determine if V. parahaemolyticus was present. Presumptive natural Vibrio counts were very low (< 2 log CFU/g) in all sampled oysters, and we did not identify any colonies with the typical morphology for V. parahaemolyticus; all colonies were either yellow, green-black, irregularly shaped, or smaller than expected for V. parhaemolyticus.

Ln 424: Such mortality is not expected in such a short time, any hypothesis regarding the factors involved? One can question the results of this trial since the other species were in the same tank. Were mortalities also observed in the other two species?

- We were also surprised at the mortality that occurred in C. gigas in this trial, as we had not observed similar mortality in previous trials. In C. sikamea and C. virginica used for this trial, the only mortality that occurred was a single C. virginica individual from the non-pathogenic V. parahaemolyticus treatment. We hypothesize that the mortality of C. gigas may have been due to environmental stress (temperature, water quality, etc.) at the time and site of harvest, combined with stress from shipping to the depuration site. We have added sentences to the manuscript to elaborate on this for the readers:

“Mortality of C. gigas was not observed in previous depuration trials, and mortality of the other oyster species in this trial was limited to one C. virginica individual from the non-pathogenic V. parahaemolyticus treatment. It is possible that the high mortality of C. gigas in this trial was due to environmental conditions such as temperature and water quality at the time and location of harvest, in addition to stress from shipping to the depuration site.”

---

## [Decision Letter · Decision Letter 1]

24 Sep 2025

Pilot-scale depuration demonstrates the suitability of non-pathogenic Vibrio parahaemolyticus as a surrogate for commercial-scale validation studies

PONE-D-25-17763R1

Dear Dr. Lunda,

We’re pleased to inform you that your manuscript has been judged scientifically suitable for publication and will be formally accepted for publication once it meets all outstanding technical requirements.

Kind regards,

Mohammad Moniruzzaman

Academic Editor

PLOS ONE

Reviewer #1: The study is well-written and provides valuable data on optimizing depuration for Vibrio reduction in oysters. The experimental design is sound and the results are clearly presented. This work should be insightful for researchers and industry players. Thank you.

Reviewer #2: The authors have taken into account the reviewers' comments. As a pilot study, this is a nice contribution to the field.

---

## [Editor Report · Acceptance letter]

PONE-D-25-17763R1

PLOS ONE

Dear Dr. Lunda,

I'm pleased to inform you that your manuscript has been deemed suitable for publication in PLOS ONE. Congratulations! Your manuscript is now being handed over to our production team.

Kind regards,

on behalf of

Dr. Mohammad Moniruzzaman

Academic Editor

PLOS ONE